# Selinexor Sensitizes TRAIL-R2-Positive TNBC Cells to the Activity of TRAIL-R2xCD3 Bispecific Antibody

**DOI:** 10.3390/cells9102231

**Published:** 2020-10-02

**Authors:** Silvia Martini, Mariangela Figini, Aurora Croce, Barbara Frigerio, Marzia Pennati, Alessandro Massimo Gianni, Cinzia De Marco, Maria Grazia Daidone, Christian Argueta, Yosef Landesman, Nadia Zaffaroni, Alessandro Satta

**Affiliations:** 1Molecular Pharmacology Unit, Department of Applied Research and Technical Development, Fondazione IRCCS Istituto Nazionale dei Tumori, 20133 Milan, Italy; silvia.martini@istitutotumori.mi.it (S.M.); Aury591@gmail.com (A.C.); marzia_pennati@hotmail.com (M.P.); 2Biomarkers Unit, Department of Applied Research and Technical Development, Fondazione IRCCS Istituto Nazionale dei Tumori, 20133 Milan, Italy; mariangela.figini@istitutotumori.mi.it (M.F.); Barbara.Frigerio@istitutotumori.mi.it (B.F.); cinzia.demarco@istitutotumori.mi.it (C.D.M.); mariagrazia.daidone@istitutotumori.mi.it (M.G.D.); 3Medical Oncology C Unit, Department of Medical Oncology and Hematology, Fondazione IRCCS Istituto Nazionale dei Tumori, 20133 Milan, Italy; alessandro.gianni@unimi.it; 4Karyopharm Therapeutics, Newton, MA 02459, USA; cargueta@karyopharm.com (C.A.); ylandesman@karyopharm.com (Y.L.)

**Keywords:** TRAIL-R2, bispecific antibody, selinexor, survivin, T cells

## Abstract

Triple-negative breast cancer (TNBC) is an aggressive disease with poor prognosis and limited therapeutic options. Recent advances in the immunotherapy field have enabled the development of new treatment strategies, among which the use of bispecific antibodies (BsAbs), able to redirect T cells against tumors, has shown promising results. In particular, a BsAb that uses TNF-related apoptosis-inducing ligand receptor 2 (TRAIL-R2) as a target was constructed and demonstrated good results in redirecting CD3^+^ T cells to kill TRAIL-R2-expressing TNBC cells. In the present study, we investigated whether treatment with selinexor, a selective inhibitor of nuclear export (SINE) targeting exportin-1/chromosome maintenance protein 1 (XPO1/CRM1), could potentiate the antitumor activity of this BsAb. In combination experiments, we found that selinexor-exposed TNBC cells exhibited greater growth inhibition when treated with the TRAIL-R2xCD3 BsAb than that expected by simple additivity. Similarly, the apoptosis rate in selinexor/TRAIL-R2xCD3 BsAb-treated TNBC cells was significantly higher than that observed after exposure to either single agent. Together, our results suggest that the combination of selinexor and TRAIL-R2xCD3 BsAb can be a viable anticancer strategy and indicate this treatment as a promising therapeutic option for TNBC patients.

## 1. Introduction

Despite ongoing surveillance by T cells and other components of the immune system, tumors develop in the presence of an intact immune system. During the initial phases of cancer development, the immune surveillance is intact and cells of the innate and adaptive immune system destroy neoplastic cells, and the long-winded ongoing campaign between the immune system and cancer cells establishes a dynamic equilibrium. However, the constant selective pressure by the immune system can promote genetic and epigenetic instability of tumor cells which eventually gives rise to variants escaping from immune surveillance [1]. The possibility to revert this status and use the adaptive arm of the immune system to fight cancer cells has been widely studied in the last decades. In particular, two immunotherapy approaches showed excellent clinical results: the use of antibodies recognizing immune checkpoints that could reactivate the immune system [2] and the use of bispecific antibodies (BsAbs) or chimeric antigen receptors (CARs) which furnish to T cells the ability to recognize tumor cells, independently of T cell receptor (TCR) specificity [3,4].

The efficiency of BsAbs that simultaneously recognize an antigen on the cancer cells and an activating receptor on the surface of immune effector cells depends on the choice of a suitable target on tumor cells. Among different tumor-associated antigens (TAA), we have previously demonstrated that the death receptor TNF-related apoptosis-inducing ligand receptor 2 (TRAIL-R2) could be used to efficiently retarget T cells and use their cytotoxic armamentarium against tumors [5]. In fact, the TRAIL-R2xCD3 BsAb that we had developed, despite poor ability to activate the extrinsic apoptotic pathway as well when multimerized by crosslinkers, was able to efficiently redirect activated T-cells and led to killing of TRAIL-R2^+^ tumor cells by perforin and granzyme. Furthermore, this BsAb demonstrated a wide range of activity against tumors of different origins including triple-negative breast carcinoma (TNBC) [6]. Although antitumor response was efficient against all tested tumor cell lines, the activity of the BsAb could be further improved by combining it with other molecules able to sensitize cells by acting on the TRAIL-R2 pathway.

It has been reported that the sensitization to TRAIL-induced cytotoxicity could be improved by interfering with the control of cell cycle or by inhibiting molecules involved as negative controllers of the apoptotic pathway [7]. Downregulation of survivin, a member of inhibitors of apoptosis protein (IAP) family, in tumor cell cytoplasm through either nonselective inhibitors [8,9,10,11,12,13] or small interfering RNAs (siRNAs) [14] sensitizes tumor cells to sTRAIL-induced apoptosis. In addition, it was demonstrated that the suppression of survivin sensitizes the cells to treatment with CD34^+^ cells genetically modified to overexpress TRAIL on their surface [15].

The export of survivin from nucleus to cytoplasm is mediated by exportin-1/chromosome maintenance protein 1 (XPO1/CRM1) [16,17]. XPO1 is overexpressed in a variety of cancer types, including TNBC, and is associated with poor prognosis in breast cancer [18,19,20,21]. Selective inhibition of XPO1/CRM1 by selinexor (XPOVIO), a small molecule approved for the treatment of relapsed or refractory multiple myeloma in 2019 and relapsed/refractory diffuse large B-cell lymphoma in 2020, and currently under clinical development in a variety of hematological and solid tumors [22], is able to downregulate cytoplasmic survivin expression thus increasing apoptosis of cancer cells [23,24].

Starting from these considerations, our study was undertaken to investigate the in vitro therapeutic potential of combining selinexor and a BsAb that is able to retarget peripheral blood lymphocytes (PBLs) to increase the killing of TRAIL-R2-positive TNBC cancer cells.

## 2. Materials and Methods

### 2.1. Cell Lines and Drugs

Human TNBC established cell lines SUM-159, MDA-MB-231 (TRAIL-R2-positive), and MDA-MB-468 (TRAIL-R2-negative) were purchased from American Type Culture Collection (ATCC, Manassas, VA, USA) and cultivated in D-MEM-F12 medium (Lonza, Basel, Switzerland) supplemented with 10% FBS. SUM-159 TRAIL-R2 knockdown clones were cultured supplementing the culture medium with 1 μg/mL puromycin (Santa Cruz Biotechnology Inc., Dallas, TX, USA). The in-house generated TNBC primary culture MS-186 was cultured in the MammoCult Human Medium Kit (Stemcell Technologies, Vancouver, BC, Canada) supplemented with heparin solution (4 µg/mL) and hydrocortisone (0.48 µg/mL). The MCF10A human breast epithelial cell line was used as a normal breast cell model and cultivated in MEBM medium (Lonza, Basel, Switzerland) supplemented with 10 ng/mL EGF, 10 ng/mL FGF, 600 U/L heparin, and B-27 Supplement (Thermo Fisher Scientific, Waltham, MA, USA). Cells were grown at 37 °C in a humidified atmosphere containing 5% CO_2_. All experiments were performed with cells that were cultured a maximum of 12 passages after thawing. To ensure the reproducibility of experiments, cells were genotyped using the Promega StemElite ID System according to ATCC guidelines. The absence of mycoplasma was screened using the PCR Mycoplasma Test Kit I/C (PromoCell, Heidelberg, Germany) according to manufacturer’s protocol. PBLs were isolated from donors’ buffy coats, provided by the Immunohematology and Transfusion Medicine Unit of Fondazione IRCCS, Istituto Nazionale dei Tumori, di Milano, after signing informed consent, and cultured as described [6]. Selinexor (XPOVIO; Karyopharm Therapeutics Inc., Newton, MA, USA) was dissolved in dimethyl sulfoxide (DMSO; Sigma-Aldrich, St. Louis, MO, USA). The bispecific antibody TRAIL-R2xCD3 was constructed, produced, purified, and characterized as previously described [5].

### 2.2. Tumor Cell Growth Inhibition and Drug Interaction Analysis

Tumor cell growth inhibition was evaluated with the CellTiter 96 AQueous One Solution Cell Proliferation Assay (MTS, #G3580; Promega, Madison, WI, USA). SUM-159 cells (1.5 × 10^3^/well), MDA-MB-468 cells (4.5 × 10^3^/well), MDA-MB-231 cells (3.5 × 10^3^/well), MS-186 cells (6 × 10^3^/well), and MCF10A cells (6 × 10^3^/well) were plated in 96-well flat-bottom microtiter plates (EuroClone SPA, Milan, Italy) and cultured for 24 h in the appropriated medium. Cells were then treated with increasing concentrations of selinexor for 24–72 h. All experiments were performed three times. The IC_50_ value was defined as the concentration of a drug inhibiting cell growth by 50%. Control cells received vehicle alone (0.01% DMSO).

In a first set of combination experiments, TNBC cells pre-treated with a single dose of selinexor for 24 h were co-cultured with human unstimulated PBLs (effector to target ratio = 5:1) for 24, 48, and 72 h in the presence/absence of 0.5 μg/mL TRAIL-R2xCD3 BsAb. At the end of each treatment, 20 μL of MTS solution was added to each well and the plate was incubated for 3 h in a 5% CO_2_ incubator at 37 °C. Absorbance at 490 nm was detected using a POLARstar optima microplate reader (BMG Labtech, Ortenberg, Germany). The percentage of viable cell was calculated with respect to untreated control cells. Where indicated, before the addition of selinexor, tumor cells were incubated with specific inhibitors of caspase-9 (z-LEHD-fmk), caspase-3 (z-DEVD-fmk), and caspase-8 (z-IETD-fmk). The interaction between selinexor and BsAb + PBL treatments was evaluated according to the method of Kern et al. [25]. An R index (calculated as expected cell survival versus observed cell survival) of 1 (additive effect) or lower indicated the absence of synergism. Supra-additivity was defined as any value of R greater than unity.

In an additional set of combination experiments, TNBC cells were seeded in duplicate in 12-well plates (5 × 10^4^/well) and pre-treated with different concentrations of selinexor (0.001–1 µM) for 24 h and then co-cultured with human unstimulated PBLs (E:T ratio = 5:1) for 48 h in the presence/absence of 0.5 μg/mL TRAIL-R2xCD3 BsAb or pretreated with a fixed dose of selinexor for 24 h and then co-cultured with human unstimulated PBLs (E:T ratio = 5:1) for 48 h in the presence/absence of different concentrations of TRAIL-R2xCD3 BsAb. Before counting with cell counter (Beckman Coulter S.p.A., Milano, Italy), culture medium was removed and adherent cells were harvested using trypsin. Results were expressed as percent variation in the number of treated cells compared with the control.

The availability of results obtained with different concentrations of the two anticancer agents allowed the analysis of drug interactions with the combination index (CI) method developed by Chou–Talalay [26,27] using the CompuSyn software (Biosoft, Cambridge, UK). CI < 1 indicates synergism, CI = 1 additivity, and CI > 1 antagonism.

### 2.3. TRAIL-R2 Knockdown

SUM-159 cells were plated for 24 h in 12-well plates ( Corning Inc., New York, NY, USA) to reach 50% of confluence with 5 μg/mL polybrene (Santa Cruz Biotechnology Inc.) in complete medium. After 16 h of incubation, SUM-159 cells were transduced overnight with lentiviral particles containing a control short RNA (shRNA) construct encoding a scrambled sequence (#sc-108080; Santa Cruz Biotechnology Inc.) or a human TRAIL-R2 shRNA composed of three target-specific constructs that encode an shRNA designed to knockdown TRAIL-R2 gene (#sc-40237-V; Santa Cruz Biotechnology Inc.). Transduced cells were maintained in medium supplemented with 1 μg/mL puromycin (Santa Cruz Biotechnology Inc.) for 1 month and polyclonal cell populations were selected. TRAIL-R2 expression on cell clones was assessed both at mRNA (by qRT-PCR) and protein (by FACS analysis) levels. Three cell clones (SUM-159ΔTa, SUM-159ΔTb, and SUM-159ΔTc) expressing decreased levels of TRAIL-R2 and one control clone (ΔCTRL) were used in the study.

### 2.4. Evaluation of TRAIL-R2 Expression

TNBC cells and TRAIL-R2-silenced SUM-159 clones were labeled with a PE-conjugated anti-TRAIL-R2 antibody (R&D System, Minneapolis, MN, USA) or an isotype control (IgG_2b_; Beckton Dickinson, Franklin Lakes, NJ, USA); fluorescence intensity was analyzed with a FACSCanto flow cytometry system (Beckton Dickinson, Franklin Lakes, NJ, USA) and FlowJo software (TreeStar Inc., Ashland, OR, USA). Data were expressed as differences between mean fluorescence intensity (MFI) obtained with the specific anti-TRAIL-R2 antibody and that of isotype control.

### 2.5. Analysis of Apoptosis Induction

To evaluate the ability of selinexor and TRAIL-R2xCD3 BsAb to induce apoptosis, SUM-159 (control and TRAIL-R2-silenced clones) and MDA-MB-468 cell lines were treated as described above. After treatment, adherent cells were pooled with detached cells and briefly subjected to annexin V/propidium iodide (PI) staining using FITC Annexin V Apoptosis Detection Kit I ( Beckton Dickinson, Franklin Lakes, NJ, USA) according to the manufacturer’s protocol. Both early (annexin V^+^/PI^−^) and late (annexin V^+/^PI^+^) apoptotic cells were included in cell death determinations. At the end of incubation, samples were analyzed by a BD Accuri C6 flow cytometer (Beckton Dickinson, Franklin Lakes, NJ, USA).

The catalytic activity of caspase-9, caspase-3, and caspase-8 was measured using APOPCYTO/caspase-9, APOPCYTO/caspase-3, and APOPCYTO/caspase-8 kits (MBL International Corp., Woburn, MA, USA), respectively, according to manufacturer’s instructions. The hydrolysis of the specific substrates was monitored by spectrofluorometry with 380-nm excitation and 460-nm emission filters. In parallel experiments, cell-specific inhibitors of caspase-9, -3, and -8 (z-LEHD, z-DEVD, and z-IETD, respectively; MBL International Corp., Woburn, MA, USA) were added before the treatment with selinexor and/or BsAb + PBLs. Results were expressed as relative fluorescence units (rfu).

### 2.6. Expression of Apoptosis-Related Proteins

Apoptosis-related proteins were quantified in whole cell lysates obtained from cells treated with selinexor alone or in combination with TRAIL-R2xCD3 BsAb + PBLs, by means of the Human Apoptosis Array Kit (R&D Systems, Minneapolis, MN, USA) according to manufacturer’s instructions. Membranes were autoradiographed and images were acquired by Imagescanner III (GE Healthcare, Milan, Italy).

### 2.7. Quantitative RT-PCR

Total RNA was isolated by RNeasy Mini Kit (QIAGEN, Hilden, Germany) according to manufacturer’s protocol and reverse transcribed using the High Capacity cDNA Reverse Transcription Kit (Applied Biosystems, Foster City, CA, USA). Quantification of survivin and TRAIL-R2 mRNA expression was assessed by quantitative RT-PCR (qRT-PCR) using the specific TaqMan Assay (Hs00153353_mL for survivin and Hs00366278_mL for TRAIL-R2; Applied Biosystems, Foster City, CA, USA). GAPDH (PN4326317) was used as a normalizer. Amplifications were realized in a final volume of 20 μL in a 96-well plate (MicroAmp, Applied Biosystems, Foster City, CA, USA) and were run on the 7900HT Fast Real-Time PCR System (Applied Biosystems, Foster City, CA, USA). Data were analyzed by SDS 2.4.1 software (Applied Biosystems, Foster City, CA, USA) and reported as relative quantity (RQ) with respect to a calibrator sample using the 2^−ΔΔCt^ method.

### 2.8. Protein Extraction and Western Blot Analysis

Cells were harvested using a scraper and lysed with RIPA buffer (Thermo Scientific Pierce, #89900). Protein samples were sonicated for 20 s and quantified through the BCA assay method ( Thermo Fisher Scientific, Waltham, MA, USA). Total cellular lysates were separated on a 4–12% NuPAGE bis-tris gel (Thermo Fisher Scientific, Waltham, MA, USA) and transferred to nitrocellulose using standard protocols. Filters were blocked in PBS-Tween 20 with 5% skim milk and incubated overnight with primary antibodies specific for TRAIL-R2 (HPA023625; Sigma-Aldrich, St. Louis, MO, USA), survivin (#ab469; Abcam, Cambridge, UK), X-linked inhibitor of apoptosis protein (XIAP) (#2042; Cell Signaling Technology, Danvers, MA, USA), and cleaved Poly (ADP-ribose) polymerase (PARP) (#5625; Cell Signaling Technology, Danvers, MA, USA). A β-actin antibody (#ab8226; Abcam, Cambridge, UK) was used to ensure equal loading of proteins.

### 2.9. Statistical Analysis

The statistical evaluation of data was done with two-tailed Student’s *t*-test. The *p*-values < 0.05 were considered statistically significant.

## 3. Results

### 3.1. Selinexor Sensitizes TRAIL-R2-Expressing TNBC Cells to Cell Death Induced by TRAIL-R2xCD3 BsAb

The expression of TRAIL-R2 was assessed using Western Blot (WB) analysis and flow cytometry in TNBC cell lines. Although to a variable extent, TRAIL-R2 expression was present in three cell lines (SUM-159 > MDA-MB-231 > MS-186), it was not detected in MDA-MB-468 cells (Figure 1a and Appendix A). Next, the cytotoxic activity of selinexor as a single agent was investigated in TRAIL-R2-expressing SUM-159, MDA-MB-231, and MS-186 cells and TRAIL-R2-negative MDA-MB-468 cells (Figure 2a–d). Tumor cells were exposed to increasing concentrations of selinexor for the indicated intervals of time and the cytotoxic activity was assessed by MTS assay. A dose- and time-dependent cell growth inhibition was observed in the different cell lines after exposure to selinexor although the sensitivity to the compound was different among cell models. Specifically, MDA-MB-468 and MDA-MB-231 cells showed a comparable and markedly higher sensitivity to the drug compared with SUM-159 cells, as indicated by the approximately 5-fold lower IC_50_ value (0.03 µM *versus* 1.00 µM and 0.90 µM, respectively) detected at 72 h (Appendix A). Interestingly, no cytotoxicity was observed in MCF10A normal breast epithelial cells after 72 h exposure to selinexor at concentrations until 10 µM (Figure 2e, Appendix A).

Co-cultures of TNBC cells with unstimulated PBLs as effector cells (E:T ratio of 5:1) were exposed to the TRAIL-R2xCD3 BsAb for different intervals of times (Figure 1b). The treatment induced a significant inhibition of tumor cell growth, measured by MTS assay and cell counting, in TRAIL-R2 highly positive SUM-159 cells after 48 and 72 h. Conversely, the growth of TRAIL-R2-negative MDA-MB-468 cells was not affected by treatment at any time considered, demonstrating the specificity of the TRAIL-R2xCD3 BsAb activity. Exposure of TNBC cells to the TRAIL-R2xCD3 BsAb in the absence of PBLs did not affect their growth (Appendix A).

In combination experiments, a 24 h pre-incubation of TNBC cells with a fixed dose of selinexor followed by treatment with the TRAIL-R2xCD3 BsAb synergistically cooperated to kill TRAIL-R2-positive SUM-159 cells, but not TRAIL-R2-negative MDA-MB-468 cells (Figure 1b). Specifically, the exposure of SUM-159 cells to selinexor in the presence of PBLs retargeted by the TRAIL-R2xCD3 BsAb induced cell growth inhibition greater than that expected by simple additivity of the effects of the two single treatments at all time points (Figure 1b). The co-culture with PBLs in the absence of the TRAIL-R2xCD3 BsAb did not modify the sensitivity of TNBC cells to selinexor, thus indicating that the addition of the BsAb is mandatory to obtain a favorable effect (Appendix A).

To better explore the interaction between selinexor and the TRAIL-R2xCD3 BsAb, SUM-159, MDA-MB-231, and MS-186 cells were pre-incubated with different selinexor doses (from 0.001 to 1.00 µM) followed by treatment with a single TRAIL-R2xCD3 BsAb concentration. A synergistic interaction between the two agents was observed at the highest selinexor doses (0.10 and 1.00 µM) in SUM-159 and MDA-MB-231, and at all tested doses in MS-186 cells (Table 1). The synergistic interaction was confirmed in SUM-159 cells pre-incubated with 0.1 µM selinexor and then exposed to TRAIL-R2xCD3 BsAb concentrations ≥5 µg/mL (Appendix A).

As an additional model to confirm the specificity of the effect induced by the TRAIL-R2xCD3 BsAb, as a single agent or in combination with selinexor, SUM-159 cells were transduced with lentiviral particles containing a construct that encodes an shRNA designed to knockdown the TRAIL-R2 gene. To rule out the possibility that attenuation of TRAIL-R2 expression was simply due to clonal divergence, three polyclonal cell populations (SUM-159ΔTa, SUM-159ΔTb, and SUM-159ΔTc) which significantly express lower levels of TRAIL-R2, both at mRNA (Figure 3a) and protein (Figure 3b,c) levels, with respect to parental and control (ΔCTRL) SUM-159 cells, were selected. Results from MTS assay showed that attenuation of TRAIL-R2 expression markedly reduced the sensitivity of SUM-159 cells to treatment with TRAIL-R2xCD3 BsAb + PBLs, alone or in combination with selinexor (Figure 3d), confirming that the presence of the TRAIL-R2 receptor is essential to induce TNBC cell growth inhibition.

### 3.2. Selinexor and TRAIL-R2xCD3 BsAb + PBL Combination Enhances Caspase-Dependent Apoptotic Cell Death in TNBC Cells

The mechanism behind the supra-additive effect observed with the selinexor and TRAIL-R2xCD3 BsAb + PBL combination treatment was studied in SUM-159 parental cells. We first analyzed by flow cytometry whether selinexor combined with TRAIL-R2xCD3 BsAb + PBLs induced an increase in cell death with respect to single treatments. An enhancement in the percentage of annexin V^+^ cells was consistently observed after combination treatment, although the difference reached statistical significance only after 48 h (Figure 4a). This observation was supported by western blot results indicating an increased level of cleaved PARP in cells exposed to the combination treatment (Figure 4b). A marked decrease in survivin and XIAP abundance was also seen in selinexor-treated cells. The protein expression was almost completely abrogated in cells exposed to the selinexor/TRAIL-R2xCD3 BsAb + PBL combination (Figure 4b). Comparable western blot results were also obtained in MDA-MB-231 and MS-186 cells (Figure 4b).

To assess whether the increased apoptotic response induced by combined treatment with selinexor and TRAIL-R2xCD3 BsAb + PBLs was sustained by caspase activation, the catalytic activities of caspase-9, caspase-3, and caspase-8 were assessed in SUM-159 cells at 48 h after individual and combined treatment. Selinexor exposure resulted in 6-, 9-, and 4-fold increase in caspase-9, caspase-3, and caspase-8 catalytic activity, respectively, compared with untreated control. TRAIL-R2xCD3 BsAb + PBL treatment induced a 5- and 8-fold increase in caspase-3 and caspase-8 catalytic activity, respectively, while no effect on caspase-9 activity was observed (Figure 4c). Notably, the selinexor and TRAIL-R2xCD3 BsAb + PBL combination resulted in a marked increase of caspase-9, caspase-8, and caspase-3 activity (15-, 11-, and 32-fold, respectively) (Figure 4c). To confirm the requirement of caspase activation for apoptosis induction by selinexor and TRAIL-R2xCD3 BsAb + PBL combined treatment, SUM-159 cells were pre-incubated with specific inhibitors of caspase-9, -3, and -8 (z-LEHD-fmk, z-DEVD-fmk, z-IETD-fmk, respectively), which were able to partially rescue cell growth (Figure 4d).

To shed light on the molecular events driving cell death induced by selinexor and TRAIL-R2xCD3 BsAb + PBLs, we assessed the levels of a panel of 35 apoptosis-related proteins. Results showed that at 48 h the combination treatment induced a reduced expression of anti-apoptotic proteins belonging to both IAP (cIAP1, cIAP2, livin, survivin, and XIAP) and Bcl-2 (Bcl-2 and Bcl-x_L_) families, which was paralleled by an increased abundance of pro-apoptotic proteins (Bad, Bax, FADD, Fas, SMAC/DIABLO, p21, and p27) (Figure 4e). Interestingly, the expression of TRAIL-R2, which is essential for the retargeting of T cells by the TRAIL-R2xCD3 BsAb, was appreciably increased after the combination treatment (Figure 4e).

Overall, these data suggest that selinexor in combination with TRAIL-R2xCD3 BsAb + PBLs induced an increased caspase-dependent apoptotic response in TNBC cells, which involved both intrinsic and extrinsic pathways.

## 4. Discussion

After showing solid advantages over traditional therapies in clinical trials, immunotherapy has become an important tool to treat a variety of cancer types. Despite the amazing clinical results, some issues need to be overcome, including the resistance to single immunotherapy regimens and the development of a variety of non-specific toxicities and side effects during the treatment [28]. To bypass these problems, the proposed strategy is combining different immunotherapeutics themselves or with other drugs [29,30].

In the present study, we proposed the combination of a BsAb able to retarget T cells to kill TRAIL-R2^-^positive cancer cells [6] with selinexor, an inhibitor of XPO1/CRM1 recently approved for the treatment of relapsed/refractory multiple myeloma and currently under clinical development in a variety of hematological and solid tumors [22]. The TRAIL-R2xCD3 BsAb mechanism of action is based on three consecutive steps, including the formation of a bridge between tumor and T cells, the activation of T cells, and the subsequent release of granzyme B and perforin that induce tumor cell killing [6]. Selinexor inhibits the export from the nucleus to cytoplasm of a variety of proteins including survivin, an anti-apoptotic protein whose downmodulation was found to sensitize cancer cells to apoptosis [15].

Previous evidence indicates that such a sensitization to apoptosis is sustained, at least in part, by the downregulation of anti-apoptotic protein of IAP family as survivin and XIAP [8,11,15]. Our data indicated that the combination of selinexor and the TRAIL-R2xCD3 BsAb could cause near total depletion of survivin and XIAP, markedly improving the activity of selinexor and BsAb alone. This feature, together with the concurrent downregulation of other anti-apoptotic molecules belonging to both IAP and Bcl-2 families and the upregulation of pro-apoptotic proteins, induced a greater apoptotic response in TRAIL-R2-positive cancer cell models, which resulted in an enhanced anticancer effect when compared with individual agents. Our results also indicated that the significant antiproliferative effect of the selinexor*/*TRAIL-R2xCD3 BsAb + PBL combination was the consequence of enhanced caspase-8 and caspase-3 activation.

Interestingly, we observed that the growth inhibitory and/or pro-apoptotic activity of the combination treatment was specific. Indeed, it was appreciable in TNBC models expressing TRAIL-R2 (SUM-159, MDA-MB-231, and MS-186), but not in negative MDA-MB-468 cells. Treatment specificity was also confirmed on SUM-159 TRAIL-R2-silenced clones. Specifically, shRNA-mediated downregulation of TRAIL-R2 expression resulted in a significantly decreased cytotoxic response of TNBC cell clones to TRAIL-R2xCD3 BsAb + PBLs (but not to selinexor) with respect to parental cells, which was dependent on cell surface expression of TRAIL-R2. This evidence further supported the relevance of receptor expression for the responsiveness of tumor cells to TRAIL-R2xCD3 BsAb and consequently to the combination treatment.

These findings, taken together with TRAIL-R2 differential expression between tumor and normal cells, strongly suggest that this receptor should be considered as a tumor-associated antigen [31,32], and corroborate previous evidence of BsAb anti-neoplastic activity [6]. Furthermore, our data form a first in vitro demonstration that the combination of the TRAIL-R2xCD3 BsAb ability to redirect the T cells cytotoxic armamentarium with the selinexor pro-apoptotic activity could be interesting as an anticancer treatment.

Overall, our data demonstrated the in vitro supra-additive cytotoxic activity of the selinexor and TRAIL-R2xCD3 BsAb combination in TRAIL-R2-positive TNBC cells. However, additional studies are needed to find the best dosage/schedule to administer the combination treatment and in vivo studies on suitable models are mandatory to explore the therapeutic potential of TRAIL-R2-expressing TNBC models.

## Figures and Tables

**Figure 1 cells-09-02231-f001:**
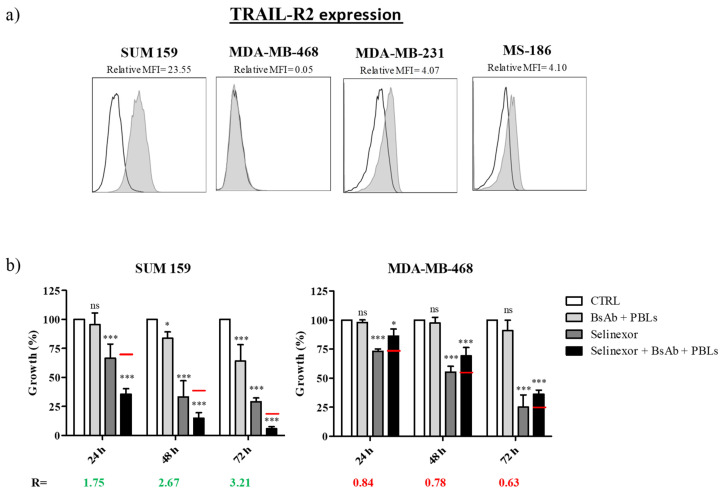
Selinexor and TRAIL-R2xCD3 bispecific antibody (BsAb)-retargeted peripheral blood lymphocytes (PBLs) cooperate to kill triple-negative breast cancer (TNBC) cells. (**a**) TRAIL-R2 expression was assessed by FACS analysis. SUM-159, MDA-MB-468, MDA-MB-231, and MS-186 cells were labeled with a commercial Phycoerythrin (PE)-conjugated anti-TRAIL-R2 mAb (grey peak); an isotype antibody was used as negative control (empty peak). (**b**) SUM-159 and MDA-MB-468 cells were exposed for 24 h to selinexor (1.0 µM and 0.2 µM, respectively) and then treated with 0.5 μg/mL TRAIL-R2xCD3 BsAb + PBLs (E:T ratio = 5:1) for additional 24, 48, and 72 h. The cytotoxic effect of individual and combined treatments was assessed by MTS assay at the indicated time points. Data are expressed as percentage values of growth in treated cells compared to control (cells exposed to 0.01% dimethyl sulfoxide (DMSO)). Bars represent the mean ± SD of three independent experiments. Red lines represent the expected additive effect of the combination, calculated as the product of the effects of the individual drugs, according to the method of Kern et al. [25]. *** *p* < 0.001, * *p* < 0.05; ns, not significant.

**Figure 2 cells-09-02231-f002:**
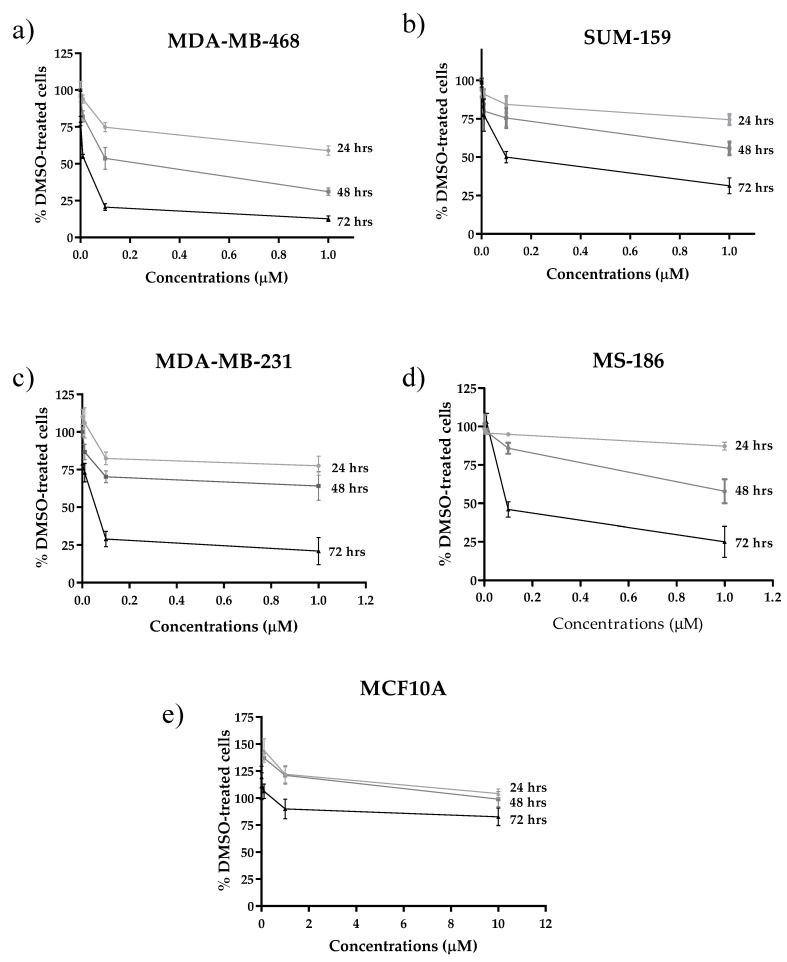
Cytotoxic activity of selinexor in TNBC and normal breast epithelial cell lines. (**a**) SUM-159, (**b**) MDA-MB-468, (**c**) MDA-MB-231, (**d**) MS-186, and (**e**) MCF10A cells were treated for 24, 48, and 72 h with increasing concentrations of selinexor and the cytotoxic activity was assessed by means of MTS assay. Data are expressed as mean ± SD of at least three independent experiments.

**Figure 3 cells-09-02231-f003:**
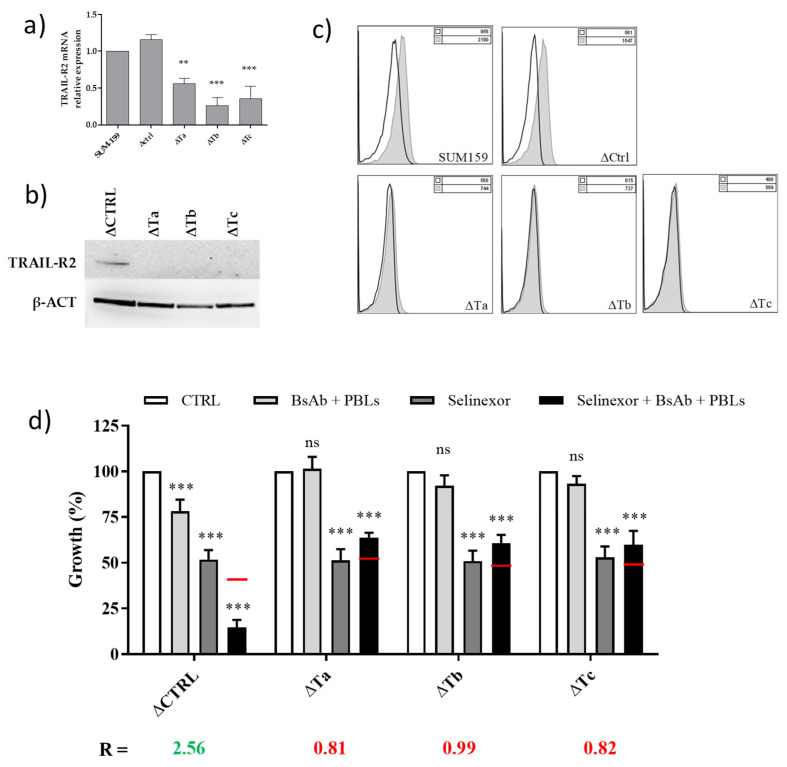
Cytotoxic effect of selinexor and TRAIL-R2xCD3 BsAb combination in SUM-159 TRAIL-R2-silenced clones. (**a**) qRT-PCR analysis of TRAIL-R2 mRNA expression levels and (**b**) western immunoblotting analysis of TRAIL-R2 protein after silencing of SUM-159 cells with lentiviral particles containing three human TRAIL-R2-specific (ΔTa, ΔTb, and ΔTc) small interfering RNA (siRNA) sequences. An empty lentiviral vector was used as negative control (ΔCTRL). Data in (a) are reported as log10-transformed relative quantity (RQ) with respect to wild-type (WT) SUM-159 cells. *** *p* < 0.001, ** *p* < 0.01. (**c**) Membrane surface expression of receptor TRAIL-R2 in SUM-159-silenced clones. TRAIL-R2 expression was assessed by flow cytometry after cell labeling with the PE-conjugated anti-TRAIL-R2 antibody (grey peak); an isotype antibody was used as negative control (empty peak). (**d**) Cytotoxic effect of treatment with selinexor and TRAIL-R2xCD3 BsAb + PBLs on SUM-159 silenced clones. Cells were exposed for 24 h to 0.1 µM selinexor and then for 48 h with 0.5 µg/mL TRAIL-R2xCD3 BsAb + PBLs (E:T ratio = 5:1). Cytotoxic effect was assessed by MTS assay. Data are expressed as percentage values of growth in treated cells compared to cells exposed to 0.01% DMSO (CTRL) and represent mean values ± SD of at least three independent experiments. *** *p* < 0.001; ns, not significant. Red lines in combination treatment bar represent the expected additive effect of the combination, calculated as the product of the effects of the individual drugs, according to the method of Kern et al. [25].

**Figure 4 cells-09-02231-f004:**
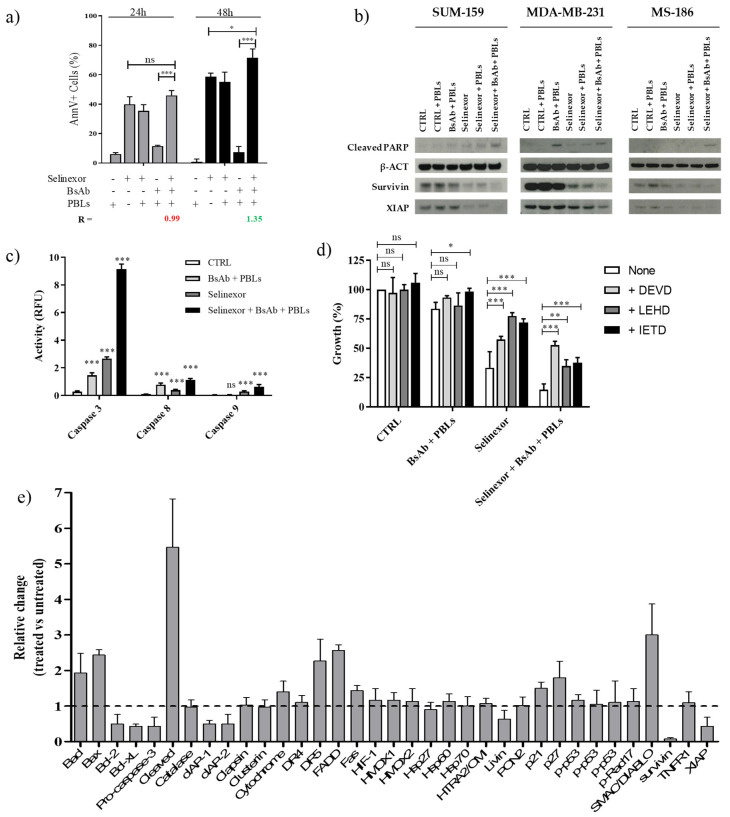
Induction of apoptosis in TNBC cells treated with the selinexor and TRAIL-R2xCD3 BsAb + PBL combination. (**a**) Induction of apoptosis was evaluated by flow cytometry as the presence of annexin V-positive cells (annexin V^+^/propidium iodide (PI)^–^ plus annexin V^+^/PI^+^ cells). Cells were exposed for 24 h to 0.1 µM selinexor followed by 24 h and 48 h treatment with 0.5 µg/mL TRAIL-R2xCD3 BsAb + PBLs (E:T ratio = 5:1). Data represent mean ± SD of at least three independent experiments. *** *p* < 0.001; * *p* < 0.05; ns, not significant. (**b**) Representative western immunoblotting showing the effect induced by individual and combined treatments on the expression of cleaved PARP, survivin, and XIAP. (**c**) Assessment of caspase-3, caspase-9, and caspase-8 catalytic activity in SUM-159 cells exposed to 0.1 µM selinexor and 0.5 µg/mL TRAIL-R2xCD3 BsAb + PBLs (E:T ratio = 5:1), alone or in combination. Data are expressed as relative fluorescence units and bars represent the mean values ± SD of at least three independent experiments. *** *p* < 0.001; ns, not significant versus untreated. (**d**) Cytotoxic effect of selinexor and TRAIL-R2xCD3 BsAb + PBLs, alone and in combination, in the presence of specific inhibitors of caspase-9, -3, and -8 (z-LEHD-fmk, z-DEVD-fmk, and z-IETD-fmk, respectively). Data are expressed as percentage values of growth in treated cells compared with cells exposed to 0.01% DMSO (CTRL), and represent mean values ± SD of at least three independent experiments. *** *p* < 0.001; ** *p* < 0.01; * *p* < 0.05; ns, not significant. (**e**) Relative levels of apoptosis-related proteins in SUM-159 cells exposed to combination of selinexor and TRAIL-R2xCD3 BsAb + PBLs. Graph shows levels of a panel of 35 apoptosis-related proteins assessed in SUM-159 cells at 48 h using the Proteome Profiler Human Apoptosis Array Kit. Data are reported as relative change of protein expression in treated compared to untreated cells and represents mean values ± SD of at least three independent experiments.

**Table 1 cells-09-02231-t001:** Combination index values for the selinexor–TRAIL-R2xCD3 BsAb combined treatment in TNBC cell lines.

SUM-159		MDA-MB-231		MS-186	
Selinexor/BsAb (µM/µg/mL)	CI ^a^	Selinexor/BsAb (µM/µg/mL)	CI ^a^	Selinexor/BsAb (µM/µg/mL)	CI ^a^
1/0.5	**0.30**	1/0.5	**0.15**	1/0.5	**0.54**
0.1/0.5	**0.21**	0.1/0.5	**0.42**	0.1/0.5	**0.34**
0.01/0.5	6.70	0.01/0.5	4.00	0.01/0.5	**0.10**
0.001/0.5	2.90	0.001/0.5	14.94	0.001/0.5	**0.05**

^a^ Combination index (CI) as determined by the median effect method using CompuSyn Software: CI < 1 indicates synergy, CI = 1 indicates additivity, and CI > 1 indicates antagonism.

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
