# Peer review of "Selinexor Sensitizes TRAIL-R2-Positive TNBC Cells to the Activity of TRAIL-R2xCD3 Bispecific Antibody"

_cells, 2020, doi:10.3390/cells9102231_

Round 1

Reviewer 1 Report

The authors have either included additional experiments or provided additional information in the revised manuscript. They also justified their conclusions based on their results. The quality of the manuscript has been improved. However, the lack of in vivo animal experiment reduced the overall significance of the study. 

Author Response

Comments and suggestions:

The authors have either included additional experiments or provided additional information in the revised manuscript. They also justified their conclusions based on their results. The quality of the manuscript has been improved. However, the lack of in vivo animal experiment reduced the overall significance of the study. 

We thank the reviewer for the positive comments. As previously highlighted in the Discussion section of the revised manuscript, we are aware that an in vivo experiment would be mandatory to assess the future clinical potential of the combined treatment; in spite of that our data strongly demonstrate the in vitro efficacy of the drugs combination which is the main aim of the study.

Reviewer 2 Report

“Selinexor sensitizes TRAIL-R2 positive TNBC cells to the activity of TRAIL-R2xCD3 bispecific antibody, presented by Martini and colleagues describes their latest efforts to combine a DR5/T-cell receptor crosslinking biologic with a nuclear export inhibitor to increase treatment efficacy in TNBC. The biologic has been characterized extensively in their prior report, has shown to exhibit affinity for both human DR5 and the T cell receptor and to induce dose-dependent target cell death in vitro. The addition of selinexor turned out to sensitize the cell death mechanism in a synergistic fashion likely via reduction of survivin expression.

In general, the paper is well written but several experimental details warrant clarification.

Major concerns:

  1. In the Method section it has been stated that the functional growth inhibition assays were performed in the presence and in the absence of PBLs (Fig. 1b). However, data to support this claim were not presented. In this regard, it would be highly informative for the reader to present data on the functional consequences of single-agent treatment arms, such as BsAb alone as well as the impact of PBLs only in the presence and absence of selinexor (no BsAb), three vital controls that were missing in Figure 1b.
  2. Given the high E:T ratio (5:1), how did the authors address the treatment effects specifically on the actual tumor cells versus a mix of tumor cells and the PBLs (MTS assay)? Similar considerations apply to the analyses and data interpretation of Fig. 3d and Fig. 4.
  3. While the contribution of the T-cell arm to the highly complex overall killing mechanism has been speculated about (a mix of death receptor activation, down modulation of anti-apoptotic intracellular molecules via pathway sensitizer selinexor, immune cell activation via release of granzyme B and perforin), the activation status of the T cells has not at all been demonstrated in the current work but should be presented.

Minor concerns:

  1. To define DR5 as a tumor association antigen (Abstract), is somewhat strong of a statement and should be rephrased, given that the extrinsic death pathway is not strictly cancer specific.
  2. The 24h and 48h time points for three cell lines are missing (Fig. 2c, d and e).
  3. The quality of Fig. 4 is quite poor (legends and axes labels are not readable). Please provide high-resolution panels for this figure.

Author Response

Major concerns:

In the Method section it has been stated that the functional growth inhibition assays were performed in the presence and in the absence of PBLs (Fig. 1b). However, data to support this claim were not presented. In this regard, it would be highly informative for the reader to present data on the functional consequences of single-agent treatment arms, such as BsAb alone as well as the impact of PBLs only in the presence and absence of selinexor (no BsAb), three vital controls that were missing in Figure 1b.

Requested controls (BsAb alone; PBLs both in presence and absence of selinexor) are now included in supplementary figure 1 and the results reported in the manuscript (lines 224-225 and 231-233).

Given the high E:T ratio (5:1), how did the authors address the treatment effects specifically on the actual tumor cells versus a mix of tumor cells and the PBLs (MTS assay)? Similar considerations apply to the analyses and data interpretation of Fig. 3d and Fig. 4.

We apologize for the mistake but we realized that in the Material and Method section of the manuscript a sentence was lacking and we added it at lines 127-134. In the second set of experiments, cells viability was assessed using a different assay in comparison to MTS. In these experiments, dead cells and PBLs were washed away before cells viability counts. In all cases the results were comparable with that obtained using MTS, so we can assume that, in all the experiments, PBLs did not influence cell viability counts after treatment.

While the contribution of the T-cell arm to the highly complex overall killing mechanism has been speculated about (a mix of death receptor activation, down modulation of anti-apoptotic intracellular molecules via pathway sensitizer selinexor, immune cell activation via release of granzyme B and perforin), the activation status of the T cells has not at all been demonstrated in the current work but should be presented.

The mechanism of action (MoA) of our bispecific antibody was extensively described in our previous published manuscript (Satta et al 2019). Briefly, the three steps by with it acts are: A) formation of the bridge between tumor and T cells B) activation of the redirected T cell C) release of granzyme B and perforin that induce the tumor cells killing. We have highlighted this feature in the Discussion section at lines 362-369. Furthermore, treatment with selinexor and selinexor+ PBLs show the same cytotoxic effect on TNBC cells, clearly indicating that PBLs were not stimulated by selinexor alone. For these reasons, we think that further experiments are dispensable.

Minor concerns:

To define DR5 as a tumor association antigen (Abstract), is somewhat strong of a statement and should be rephrased, given that the extrinsic death pathway is not strictly cancer specific.

The statement was rephrased and “tumor associated antigen” at line 29 was changed to “target”.

The 24h and 48h time points for three cell lines are missing (Fig. 2c, d and e).

The curves of the 24h and 48h treatments with selinexor were now added to fig 2c, d and e.

The quality of Fig. 4 is quite poor (legends and axes labels are not readable). Please provide high-resolution panels for this figure.

The original Fig.4 was substituted with a higher resolution one.

Round 2

Reviewer 2 Report

The authors have addressed the critiques adequately. The only remaining issue regards Suppl. Fig. S2, in which the statistical analysis is presented in the legend but is not part of the figure (*P<0.05 as well as NS).

Author Response

The authors have addressed the critiques adequately. The only remaining issue regards Suppl. Fig. S2, in which the statistical analysis is presented in the legend but is not part of the figure (*P<0.05 as well as NS).

The statistical analysis is not present in the figure and for this reason is now removed from the legend.